# Improving the Tribological Performance of MAO Coatings by Using a Stable Sol Electrolyte Mixed with Cellulose Additive

**DOI:** 10.3390/ma12244226

**Published:** 2019-12-16

**Authors:** Wei Song, Bailing Jiang, Dongdong Ji

**Affiliations:** 1Faculty of Materials Science and Engineering, XI’AN University of Technology, NO.5 South Jinhua Road, Xi’an 710048, Shaanxi, China; song78wei@163.com (W.S.); jdd3141592654@163.com (D.J.); 2School of biological and Chemical Engineering, Nanyang Institute of Technology, NO.80 Changjiang Road, Nanyang 473004, He’nan, China

**Keywords:** cellulose, tribological performance, stability, MAO (micro-arc oxidation) coating, self-lubricating

## Abstract

In this study, micro-arc oxidation (MAO) of aluminum 6061 alloy was carried out within a silicate base electrolyte containing 0.75 g/L of cellulose, and the tribological properties of the coating were investigated. The as-prepared coating was detected by Fourier Transform Infrared Spectroscopy (FTIR), X-ray diffraction (XRD), a scanning electron microscope (SEM) and an energy-dispersive spectrometer (EDS), respectively. The results suggested that cellulose filled in the microcracks and micropores, or it existed by cross-linking with Al^3+^. In addition, it was found that the cellulose had little effect on the coating hardness. However, the thickness and roughness of the coating were improved with the increase in cellulose concentration. Moreover, the ball-on-disk test showed that the friction coefficient, weight loss and wear rate of the MAO coating decreased with the increase in cellulose concentration. Further, the performances of the coatings obtained in the same electrolyte, under different preserved storage periods, were compared, revealing that the cellulose was uniformly dispersed in the electrolyte and improved the tribological properties of the MAO coating within 30 days.

## 1. Introduction

Aluminum alloys are characterized by their excellent castability, high specific strength and low thermal expansion coefficient [1]. As a result, they have aroused increasing interest in the automobile industry, as well as having aerospace structural and military applications [2,3,4]. Nonetheless, aluminum alloys are associated with poor tribological performances, since the friction coefficient is as high as 0.5–0.8 under dry friction conditions [5]. Typically, aluminum alloys exhibit poor tribological performances when they come into contact with other metal materials. This is ascribed to severe adhesive wear, plastic deformation and metallic wear [6]. Consequently, surface modification approaches are indispensable when it comes to enhancing the tribological performances of aluminum alloys. 

A variety of surface treatment techniques are available at present, such as the electrochemical approach [7], electroless deposition [8], chemical surface conversion [9], deposition from the gas-phase [10], laser surface alloying [11] and organic polymer coating [12]. Micro-arc oxidation (MAO), also referred to as plasma electrolytic oxidation, sparks anodization or micro-plasma discharge oxidation [13,14] and emerges as a unique technique to produce hard and thick ceramic oxide coatings on diverse Al [15], Mg [16] and Ti [17] alloys. Noteworthily, coatings synthesized according to the MAO process exhibit superior mechanical properties, including excellent adhesive strength [18], high micro-hardness [19], and high thermal conductivity [20] compared with those obtained through other methods. MAO coatings offer several advantages over other coatings. MAO coatings are very stable and hard, which means they can be used at high temperatures. MAO treatment can significantly enhance the surface properties of Mg, Al, Ti and their alloys. For instance, MAO coatings exhibit better anti-wear and anti-corrosion performances than other chemical conversion layers. In addition, the pores and cracks generated in MAO coatings during micro-arc discharges can help relieve the residual stress of the coating. Thus, MAO coatings are promising for the corrosion protection of aluminum and magnesium [21], the wear resistance of light metals and their alloys [22], and the improved biofunctionality of titanium [23]. In addition, the composition, structure, and properties of coatings produced by the MAO process depend on various parameters, among which chemical composition and electrolyte concentration are the most important [24,25,26]. The microstructures and properties of diverse composites have also been extensively investigated in plenty of reviews and books [27,28,29,30]. 

Generally, composites are added into the electrolyte to improve the tribological performances of aluminum alloys, since they are able to compact the coating by filling in the microcracks and micropores of the MAO coating [31,32], sealing the surface or reacting with the aluminum ion as the coating forming matter [33,34].

However, these techniques are linked with certain shortcomings [35,36]:The carbide and nitride oxide that must be mixed into the metal matrices are so hard and brittle that they may be broken in the course of mixing or in the consolidation processes.The additive may not be uniformly dispersed into the electrolyte.A chemical reaction between the metal matrix and the coating may occur during the exposure to elevating temperature, which leads to poor mechanical properties of the composites.The particle sizes are typically in tens to hundreds of microns, which considerably reduces the ductility and toughness, as well as ineffectively utilizing the strength and stiffness of the reinforcement.The electrolyte is unstable and cannot be used in actual industrial production.

Cellulose is different from the above components because it is a polymer compound that contains multiple hydroxyl groups and experiences limited swelling under alkaline conditions, resulting in the formation of a stable uniform sol electrolyte [37].

Figure 1 illustrates that the anti-friction mechanism of cellulose improves the MAO coating. As a polymer, cellulose possesses favorable self-lubricity and plastic deformation ability. The possible mechanisms by which cellulose typically prompts the tribological performance are explained below.

First, the MAO coating can be decreased depending on the self-lubrication ability of the cellulose. Secondly, the propagation of microcracks and micropores generated by thermal stress in the coating can be inhibited based on the plastic deformation capacity. Thirdly, the cellulose fills in the microcracks and micropores. As a result, the coating compactness is increased when it forms the complex with the aluminum ion.

It is well known that the coating’s tribological performance is enhanced with an increase in compactness and a decrease in friction coefficient, whereas an increase in coating toughness [38] reduces the occurrence of adhesive wear [39], which seriously affects the service life of the coating [40].

On this account, mixing cellulose into the electrolyte contributes to obtaining a stable sol solution with uniform dispersion, which is helpful for preparing a MAO coating with excellent tribological performance.

This study mainly aimed to improve the tribological performance of the MAO coating by adding cellulose into the electrolyte. Moreover, the effect of cellulose content on the tribological performance of the coating was investigated under optimized parameters; subsequently, the tribological performances of coatings obtained during different storage periods were compared in order to investigate the stability of the sol electrolyte.

## 2. Experimental Procedure

### 2.1. Materials

Aluminum 6061 alloy (AA 6061) specimens with the dimensions of 20 × 20 × 3.5 mm^3^ were used as the anodized substrates. Specifically, the alloy composition by wt % included 0.8–1.2% Mg, 0.4–0.8% Si, 0.15–0.4% Cu, and 0.04–0.35% Cr, and Al was the balance. The contents of Fe, Mn, Zn and Ti in the alloy were not higher than 0.7, 0.15, 0.25 and 0.15 wt %, respectively. Prior to the experiment, all specimens were mechanically ground using 240, 400 and 800 grit silicon carbide paper and then washed with distilled water.

### 2.2. Experiment Process

Two 300 × 300 mm^2^ AISI 321 stainless steel sheets (BENLAIMETAL, Shanghai, China) were used as the cathode. Then, the MAO process was carried out in a stirred electrolyte consisting of 15 g/L Na_2_SiO_3_ (MACKLIN, Beijing, China), 5 g/L KOH (MACKLIN, Beijing, China), and 5 g/L (NaPO_3_)_6_ (MACKLIN, Beijing, China). Except for the adjusted cellulose concentrations (0, 0.25, 0.50, 0.75, and 1 g/L), the MAO processes of specimens were carried out at 20 °C for 30 min with a DC pulse supply at the frequency of 50 kHz, the constant current density of 1 A/cm^2^, and the duty cycle of 15%. To examine the electrolyte stability, the MAO processes were conducted within the same electrolyte after various storage periods for 0, 1, 7, 14 and 30 days.

### 2.3. Characterization

The phase compositions of diverse coatings were examined through an X-ray diffractometer (XRD) (D/max-rB, RICOH, Tokyo, Japan) with a Cu Ka source, and the accelerating voltage and applied current were 40 kV and 30 mA, respectively. In addition, the radiation emitted by the sample surface was detected by a Fourier transform infrared (FT-IR) spectrometer (JASCO FT/IR-6100, JASCO, Toykp, Japan). Meanwhile, scanning electron microscopy (SEM, JSM-6700F, JEOL, Japan) was employed to observe the microstructure and morphology of the MAO coatings, whereas the element compositions on the coating were analyzed by an energy-dispersive spectrometer (EDS, Oxford, UK) combined with SEM. Further, the coating thicknesses generated under different conditions were measured using an eddy current coating thickness measurement gauge (CTG-10, Time Company, Beijing, China). To be specific, the thicknesses at 10 different sites on the coating surface were measured to calculate and record the average. Additionally, the coating roughness was tested using a roughness tester (TR-3200, Time Company, Beijing, China, vertical resolution of 0.01 µm). The micro-hardness of the coating was evaluated using the HVS-100 micro-hardness tester (TMVS-1, TIMES Group, Beijing, China) with a load of 100 g for 10 s. In addition, the tribological behaviors of the coatings were evaluated using the ball-on-disk tester (UMT-Tribolab, BRUKER, Bremen, Germany) under dry sliding conditions. Typically, balls of GCr15 with a diameter of 10 mm and a hardness of HRC 60 were used as the counterface materials. The normal load was 5 N, and the linear sliding speed was 0.01 m/s. All tests were run under the laboratory conditions (temperature of 25 °C and relative humidity of 50%) for 30 min each. The friction coefficient was recorded on a computer during each test. The wear loss was weighed using an electronic balance, and the wear rate (k) was calculated according to the following Formula (1).
(1)$$k=\frac{\pi •D[\arcsin(\frac{L_{u}}{2r})r^{2}-L_{u}(\frac{\sqrt{4r^{2}-L_{u}{}^{2}}}{4})]}{P\times S}$$
where *r* stands for the radius of the corundum ball (mm), *D* represents the diameter of the wear track (mm), *L_u_* indicates the width of the wear track (mm), *S* is the sliding distance (m), and *P* is the applied normal load (N). 

## 3. Results and Discussion

### 3.1. Thickness and Roughness of the MAO Coating

The tribological performance of the MAO coating was affected by its thickness and roughness; therefore, the impacts of cellulose content on the coating thickness and roughness were examined, as shown in Figure 2. 

It can be observed from Figure 2 that with the increase in cellulose concentration, the thickness of the MAO coating increased, whereas its roughness decreased. At the cellulose content of 0.75 g/L, the coating thickness and roughness reached 32.1 μm microns and 0.66 μm, respectively. However, further increase in the cellulose concentration showed no obvious improvement in the thickness and roughness of the MAO coating, which was mainly ascribed to the low cellulose content. Specifically, it contained the multiple-hydroxy, and a double electric layer was formed during the electrochemical process, which attracted Al^3+^ contiguous to the substrate surface [38], resulting in the increase in the MAO coating thickness. In addition, cellulose participated in the coating formation by filling in the microcracks and micropores, even cross-linking with Al^3+^ in the coating, as observed in Figure 1. Moreover, the excellent polymer plasticity contributed to reducing the quantity and size of microcracks and micropores, thus decreasing the MAO coating roughness. Nonetheless, the Al^3+^ escaping from the substrate was limited by the electrochemical parameters, and further increase in the cellulose contents showed no significant improvement of the coating thickness and roughness when most of the Al^3+^ ions were attracted by the cellulose. Additionally, the electrolyte became inhomogeneous after over 24 h of preservation, so the optimal cellulose content was determined to be 0.75 g/L. 

### 3.2. Microstructure of the MAO Coating

The surfaces and cross-section microstructures of the MAO coatings at different cellulose contents were observed through SEM. The results are shown in Figure 3 and Figure 4, respectively. As observed from Figure 3, the increase in cellulose content led to the decreased size of the microcracks and micropores, while it increased the quantity of the micropores. 

The cross-section photograph displayed in Figure 4 proves the above findings. In addition, Figure 4 also suggests that when the cellulose concentration was 0 g/L, the coating thickness was small and the adhesion between coating and substrate was poor (seen in Figure 4a). With the increase in cellulose concentration, the coating thickness increased and the adhesion between coating and substrate was enhanced (seen in Figure 3b,c,d). The possibility of adhesive wear was reduced with the increase in the bonding force between the coating and the substrate. To further investigate the coating component, EDS was carried out. The carbon contents at different sites are presented in Figure 5 and Table 1. As displayed in Table 1, the carbon element spread all over the coating, which proved that part of the cellulose filled in the microcracks and micropores, while part of it cross-linked with the Al^3+^ in the coating. In addition, the cellulose content in the micropores and microcracks was higher than it was at the other sites, indicating that they were filled in by a relatively small portion of the cellulose.

### 3.3. Phase Structure of the Coating

The crystalline phase compositions of the MAO coatings at the cellulose contents of 0, 0.25, 0.50, 0.75 and 1 g/L were analyzed by means of FTIR and XRD, respectively. The results are shown in Figure 6 and Figure 7, separately. Figure 6 illustrates the infrared absorption peaks of the MAO coatings obtained under various cellulose contents. Notably, the peaks at 3406 cm^−1^ were assigned to O–H stretching vibrations, while those at 1630 cm^−1^ corresponded to C–O stretching vibrations, and those at 838 and 648 cm^−1^ were indexed to Al–O stretching vibrations. As indicated by Figure 7, the increase in the cellulose content gave rise to the enhanced characteristic peak of the cellulose and the decreased peak intensity of the alumina. Taken together, the analyzed results of the FTIR and XRD spectra proved the presence of cellulose in the MAO coating.

### 3.4. Tribological Performances of the MAO Coatings

Figure 8 exhibits the influences of the cellulose content on the friction coefficient. Clearly, the friction coefficient was significantly reduced after the aluminum alloys were treated by the MAO technology, and it slowly decreased with the further increase in the cellulose content. In addition, the friction coefficients of most samples were maintained at fixed values when cellulose was used as the additive; however, that of the cellulose-free MAO coating was suddenly increased after 20 min. These findings revealed that the MAO surface treatment technology reduced the friction coefficient of the aluminum alloy, and the addition of cellulose into the electrolyte was beneficial to further decrease and maintain the friction coefficient for a long time. To examine the tribological properties of the MAO coatings obtained at different cellulose contents, the micro-hardness of the MAO coatings were tested by the micro-hardness tester, while the wear loss (the amount of material lost during the mechanical tests) and wear rate were determined through wear tests. Results are presented in Table 2.

According to Table 2, the micro-hardness of the coatings remained at about 1230 HV_0.1_, while the wear loss and wear rate decreased when the cellulose content was elevated from 0% to 1.0%. These results suggest that the addition of cellulose was beneficial for improving the tribological performances of the MAO coatings.

### 3.5. Stability of the Electrolyte

Apart from the favorable anti-wear performance, the electrolyte stability, especially when polymer is used as the additive, is also a crustal parameter in practical industrial production. To investigate the electrolyte stability during long-term storage, the performances of the MAO coatings (such as thickness, roughness, hardness, friction coefficient, wear loss and wear rate) obtained at different electrolyte storage periods were compared, as shown in Figure 9 and Table 3. There was no obvious difference between them, demonstrating that the electrolyte might be employed to improve the tribological performance of the MAO coating within 30 days.

## 4. Conclusions

The tribological performances of MAO coatings are improved by mixing 0.75 g/L of cellulose into the electrolyte. The thickness of the coating increases while the roughness decreases with the increase in cellulose content.The coating compositions are thereby analyzed by FTIR and XRD, which prove the presence of cellulose in the coating. Moreover, the coating microstructure is observed through SEM, which reveals that the coating has a compact structure; meanwhile, the coating compositions at micropores, microcracks and normal positions are examined through EDS, suggesting that part of the cellulose fills in the microcracks and micropores, and part of it cross-links with the Al^3+^.The tribological performances of the coatings at different cellulose concentrations are evaluated using a ball-on-disk tester under dry sliding conditions. After different storage periods, they are compared at the same electrolyte, revealing that MAO coatings with consistent quality can be produced in this electrolyte.

## Figures and Tables

**Figure 1 materials-12-04226-f001:**
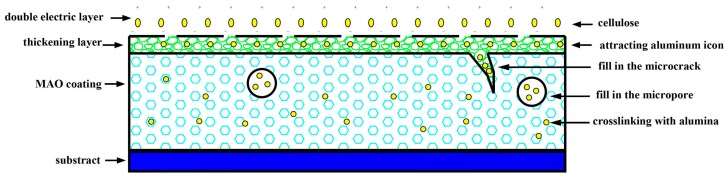
The anti-friction mechanism of cellulose improves the micro-arc oxidation (MAO) coating.

**Figure 2 materials-12-04226-f002:**
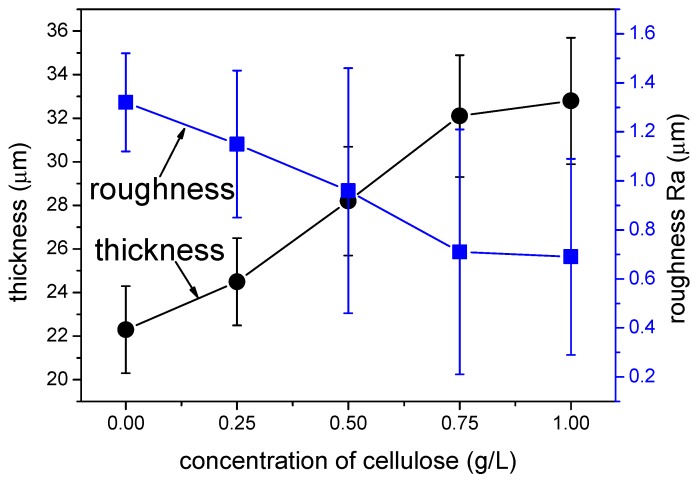
Effects of cellulose content on the thickness and roughness of the MAO coating.

**Figure 3 materials-12-04226-f003:**
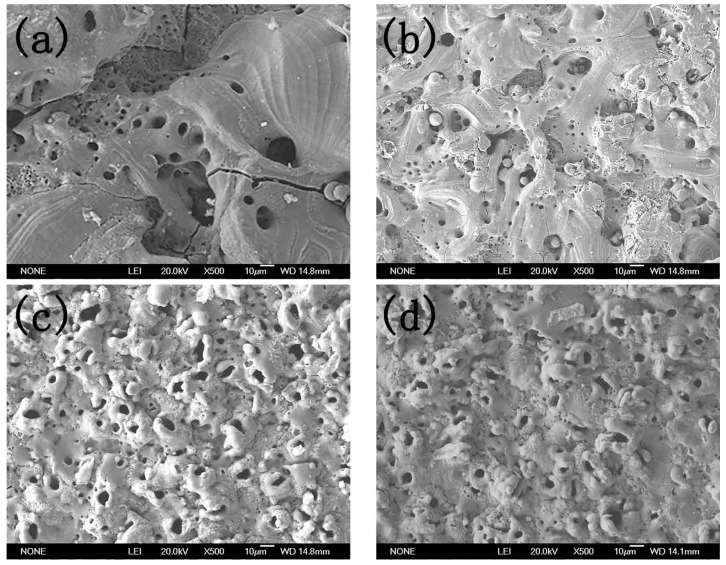
Microsurface of the MAO coating at cellulose concentrations of (**a**) 0 g/L, (**b**) 0.25 g/L, (**c**) 0.50 g/L, and (**d**) 0.75 g/L.

**Figure 4 materials-12-04226-f004:**
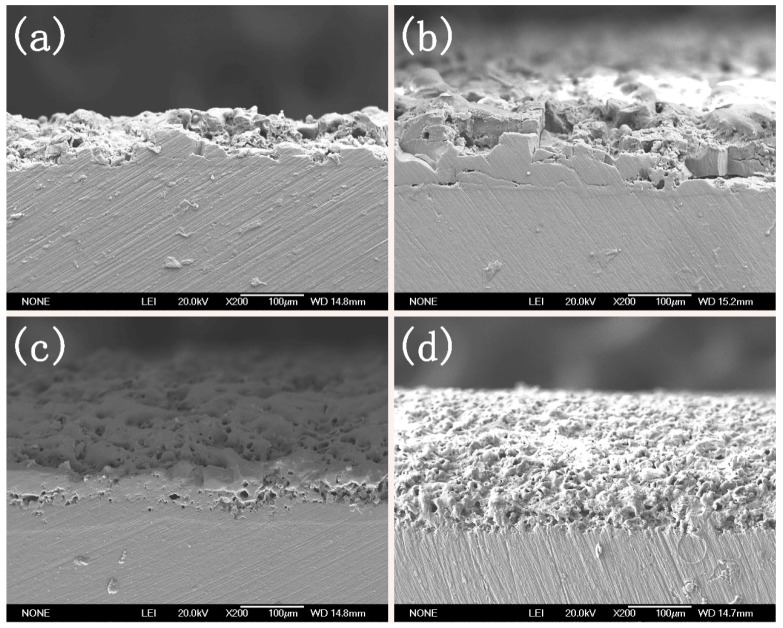
Cross-section of the MAO coating at cellulose concentrations of (**a**) 0 g/L, (**b**) 0.25 g/L, (**c**) 0.50 g/L, and (**d**) 0.75 g/L.

**Figure 5 materials-12-04226-f005:**
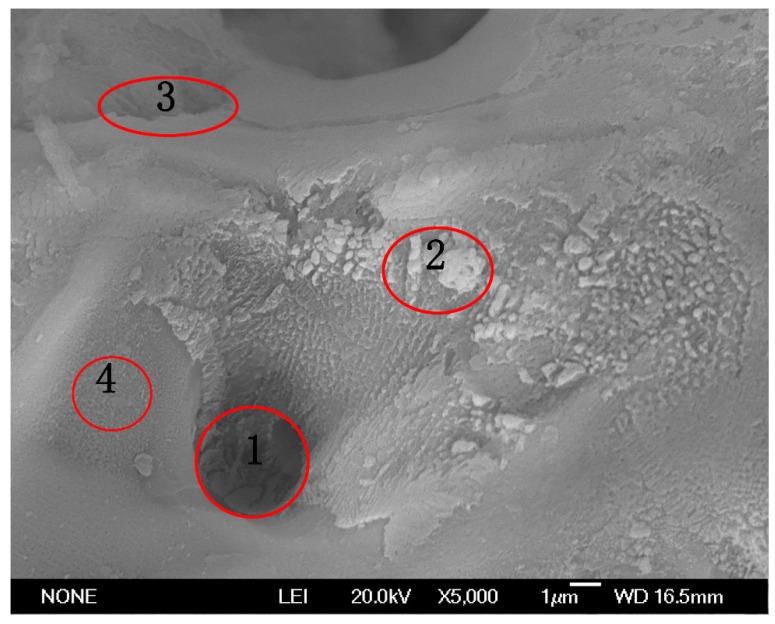
Micrograph illustrating the zone of energy-dispersive spectrometer (EDS) analysis.

**Figure 6 materials-12-04226-f006:**
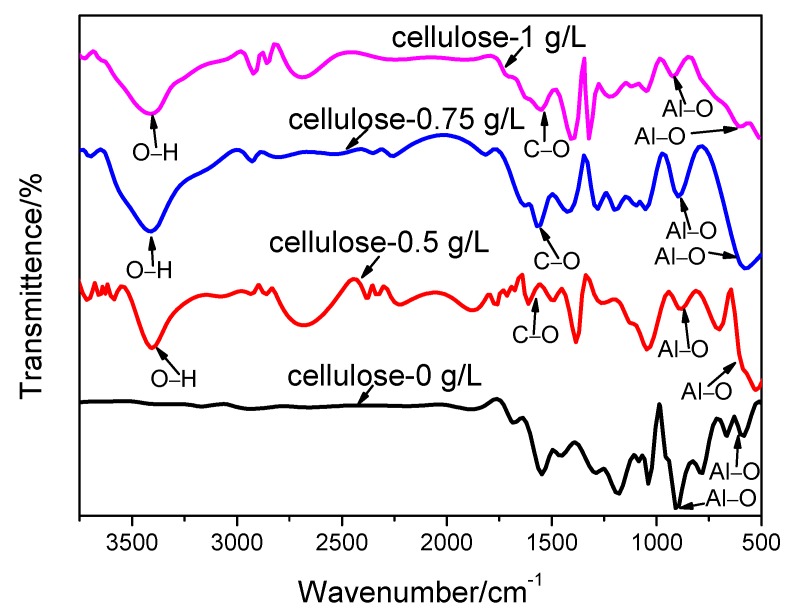
Fourier transform infrared (FT-IR) spectra of the MAO coatings obtained at different cellulose contents.

**Figure 7 materials-12-04226-f007:**
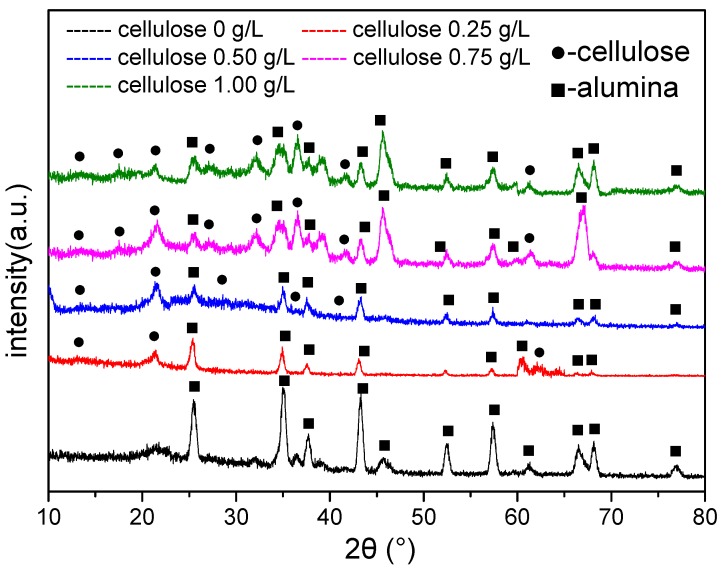
X-ray diffraction (XRD) spectra of the MAO coatings obtained at different cellulose contents.

**Figure 8 materials-12-04226-f008:**
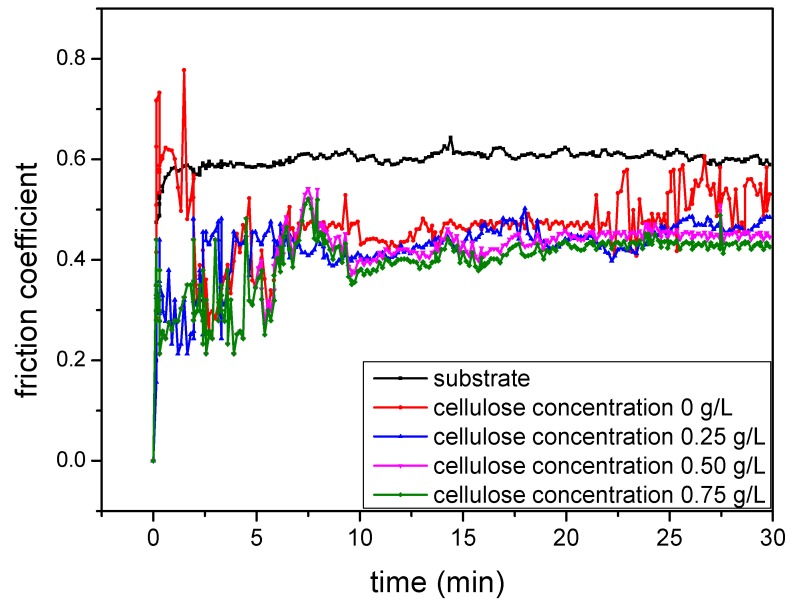
Friction coefficients of the MAO coatings obtained at different cellulose contents.

**Figure 9 materials-12-04226-f009:**
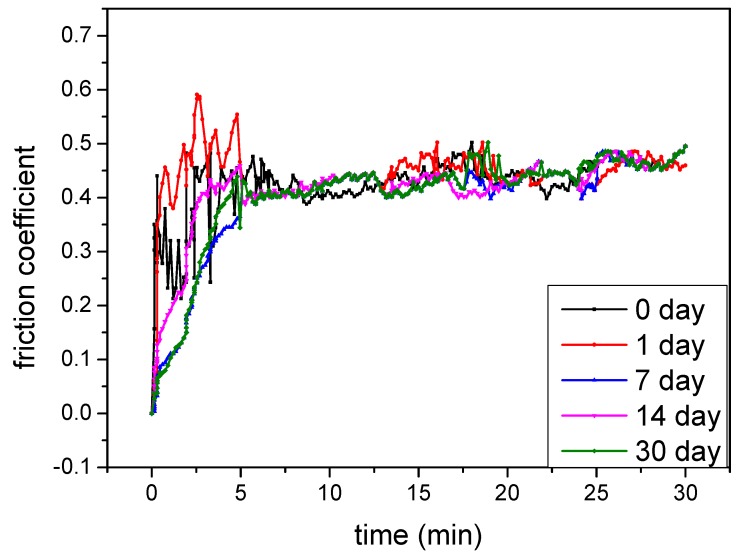
Friction coefficients of the MAO coatings under different storage periods.

**Table 1 materials-12-04226-t001:** C, O, Al element contents at different positions.

Point	C K (at. %)	O K (at. %)	Al K (at. %)
1	4.21	45.31	42.48
2	3.16	48.43	45.41
3	3.83	46.54	43.63
4	2.43	49.84	46.73

**Table 2 materials-12-04226-t002:** Hardness and tribological performances of the MAO coatings obtained at different cellulose contents.

Cellulose Concentration g/L	Micro-Hardness HV_0.1_	Weight Loss mg	Wear Track Depth µm	Wear Track Width µm	Wear Rate 10^−5^ mm^3^/N·m
0	1260	16	6.42	679.61	2.30
0.25	1230	14	4.59	537.49	1.13
0.50	1240	12	4.31	522.32	1.03
0.75	1230	11	3.54	486.26	0.84
1.00	1220	10	3.13	482.01	0.82

**Table 3 materials-12-04226-t003:** Coating performances under different storage periods.

Stability of the Electrolyte Day	Micro-Hardness HV_0.1_	Weight Loss mg	Thickness µm	Roughness µm	Wear Rate 10^−5^ mm^3^/N·m
0	1230	11	32.1	0.66	0.84
1	1230	15	32.2	0.72	0.79
7	1220	10	31.9	0.68	0.91
14	1230	11	32.1	0.69	0.89
30	1220	11	31.9	0.67	0.92

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
