# Peer review of "Improving the Tribological Performance of MAO Coatings by Using a Stable Sol Electrolyte Mixed with Cellulose Additive"

_materials, 2019, doi:10.3390/ma12244226_

Round 1

Reviewer 1 Report

Abstract

Check spelling (line 17).

Introduction

Correct reference notation (line 50).

Figure 1 is not mentioned in the text (line 60).

Results

Figure 3 should be provided with higher resolution.

Author Response

Dear Reviewer:

On behalf of my co-authors, we thank you very much for giving us an opportunity to revise our manuscript, we appreciate editor and reviewers very much for their positive and constructive comments and suggestions on our manuscript entitled“Improving The Tribological Performance of MAO Coating by Using A Stable Sol Electrolyte Mixed with Cellulose Additive”. (ID: 663081).

We have studied reviewer’s comments carefully and have made revision which marked in blue in the paper. We have tried our best to revise our manuscript according to the comments. Attached please find the revised version, which we would like to submit for your kind consideration.

We would like to express our great appreciation to you and reviewers for comments on our paper. Looking forward to hearing from you.

Thank you and best regards.

Yours sincerely,

Song Wei

E-mail:song78wei@163.com

Name: Jiang Bai-ling

E-mail: jiangbail@vip.163.com

List of Responses

Reviewer #1:

The reviewers comment:1) correct the spelling of line 17 and reference of line 50.

Answer: We are very sorry for our incorrect writing. We have made correction according to the reviewer’s comments.  

The reviewers comment: 2) figure 3 provided higher resolution.

Answer: Considering the reviewer’s suggestion, the samples were rescanned, the SEM photograph were obtained with higher resolution(×500 times).

The reviewers comment:3) figure 1 is not mentioned in the text.

Answer: We are very sorry for our ignored writing. We have made correction according to the reviewer’s comments.

Special thanks to you for your good comments.

Reviewer 2 Report

Congratulations to the authors for their good work. The following comments will help you to improve the quality of your work prior to publishing:

please add a paragraph to your introduction to explain the benefits of MAO and its applications in different fields. for instance, you can talk about the benefit of MAO coatings in biomedical applications, etc. Also, you can talk about different substrates used in MAO. the following publication and further publications could help you to do this. it is strongly recommended to use the following references:

https://link.springer.com/article/10.1007/s10853-019-03375-1

https://www.sciencedirect.com/science/article/pii/S0257897218311964

the title of your manuscript, please change Mao to MAO. you should keep the abbreviations in capital letters. add keywords after your abstract. discussion on figure 4 is not clear. please add more discussion and explain what is changed in figs a, b, c, and d. in measuring the friction coefficients, did you measure them more than once? if yes, you might need to have error bars. make the conclusion in bullet points in order to make it easier for the readers to get the summary of your work.

Author Response

Dear Reviewer:

On behalf of my co-authors, we thank you very much for giving us an opportunity to revise our manuscript, we appreciate editor and reviewers very much for their positive and constructive comments and suggestions on our manuscript entitled“Improving The Tribological Performance of MAO Coating by Using A Stable Sol Electrolyte Mixed with Cellulose Additive”. (ID: 663081).

We have studied reviewer’s comments carefully and have made revision which marked in blue in the paper. We have tried our best to revise our manuscript according to the comments. Attached please find the revised version, which we would like to submit for your kind consideration.

We would like to express our great appreciation to you and reviewers for comments on our paper. Looking forward to hearing from you.

Thank you and best regards.

Yours sincerely,

Song Wei

E-mail:song78wei@163.com

Name: Jiang Bai-ling

E-mail: jiangbail@vip.163.com

List of Responses

The reviewers comment:1) add a paragraph in the introduction to explain the benefit of MAO and its application in different fields.

Answer: We have re-written this part according to the reviewer’s suggestion and update the references.

The reviewers comment:2) correct the Mao to MAO

Answer:We are very sorry for our incorrect writing. We have made correction according to the reviewer’s comments.

The reviewers comment:3) add more discussion and explain about figure 4.

Answer:As reviewer suggested that we have re-written this part.

The reviewers comment:4) make the conclusion in bullet points in order the readers read easy.

Answer:We have re-written this part according to the reviewer’s suggestion.

The reviewers comment:5) add Key words

Answer:We are very sorry for our incorrect writing. We have made correction according to the reviewer’s comments.

The reviewers comment:6) the friction coefficient should be measured several times and indicate the error.

Answer: It is really true as reviewer suggested that the friction coefficient were measured several times, but the friction coefficient is a curve of thousands of data, which is difficult to express the error bar.   

Special thanks to you for your good comments.

Other changes:

Inspired by the experts, figure 2Effects of cellulose content on the thickness and roughness of the MAO coatingwas revised and error bar was added. we have corrected the Al3+ to Al3+ in all of our article.

We tried our best to improve the manuscript and made some changes in the manuscript. These changes will not influence the content and framework of the paper. And here we did not list the changes but marked in blue in revised paper.

We appreciate for editors/reviewers’ warm work earnestly, and hope that the correction will meet with approval.

Once again, thank you very much for your comments and suggestions.

Round 2

Reviewer 2 Report

Thank you for the good work.

Good luck!